# Using Visual Patient to Show Vital Sign Predictions, a Computer-Based Mixed Quantitative and Qualitative Simulation Study

**DOI:** 10.3390/diagnostics13203281

**Published:** 2023-10-23

**Authors:** Amos Malorgio, David Henckert, Giovanna Schweiger, Julia Braun, Kai Zacharowski, Florian J. Raimann, Florian Piekarski, Patrick Meybohm, Sebastian Hottenrott, Corinna Froehlich, Donat R. Spahn, Christoph B. Noethiger, David W. Tscholl, Tadzio R. Roche

**Affiliations:** 1Institute of Anesthesiology, University Hospital Zurich, University of Zurich, 8091 Zurich, Switzerland; amos.malorgio@usz.ch (A.M.); david.henckert@gmail.com (D.H.); giovanna.schweiger@usz.ch (G.S.); donat.spahn@swisspbm.ch (D.R.S.); christoph.noethiger@usz.ch (C.B.N.); david.tscholl@usz.ch (D.W.T.); 2Departments of Epidemiology and Biostatistics, Epidemiology, Biostatistics and Prevention Institute, University of Zurich, 8001 Zurich, Switzerland; julia.braun@uzh.ch; 3Department of Anesthesiology, Intensive Care Medicine, and Pain Therapy, University Hospital Frankfurt, Goethe University Frankfurt, 60323 Frankfurt, Germany; zacharowski@med.uni-frankfurt.de (K.Z.); raimann@med.uni-frankfurt.de (F.J.R.); florian.piekarski@ukbonn.de (F.P.); 4Department of Anesthesiology, Intensive Care, Emergency, and Pain Medicine, University Hospital Wuerzburg, 97070 Wuerzburg, Germany; meybohm_p@ukw.de (P.M.); hottenrott_s@ukw.de (S.H.); froehlich_c1@ukw.de (C.F.)

**Keywords:** avatar, machine learning, monitoring, predictive models, Visual Patient, vital sign predictions

## Abstract

Background: Machine learning can analyze vast amounts of data and make predictions for events in the future. Our group created machine learning models for vital sign predictions. To transport the information of these predictions without numbers and numerical values and make them easily usable for human caregivers, we aimed to integrate them into the Philips Visual-Patient-avatar, an avatar-based visualization of patient monitoring. Methods: We conducted a computer-based simulation study with 70 participants in 3 European university hospitals. We validated the vital sign prediction visualizations by testing their identification by anesthesiologists and intensivists. Each prediction visualization consisted of a condition (e.g., low blood pressure) and an urgency (a visual indication of the timespan in which the condition is expected to occur). To obtain qualitative user feedback, we also conducted standardized interviews and derived statements that participants later rated in an online survey. Results: The mixed logistic regression model showed 77.9% (95% CI 73.2–82.0%) correct identification of prediction visualizations (i.e., condition and urgency both correctly identified) and 93.8% (95% CI 93.7–93.8%) for conditions only (i.e., without considering urgencies). A total of 49 out of 70 participants completed the online survey. The online survey participants agreed that the prediction visualizations were fun to use (32/49, 65.3%), and that they could imagine working with them in the future (30/49, 61.2%). They also agreed that identifying the urgencies was difficult (32/49, 65.3%). Conclusions: This study found that care providers correctly identified >90% of the conditions (i.e., without considering urgencies). The accuracy of identification decreased when considering urgencies in addition to conditions. Therefore, in future development of the technology, we will focus on either only displaying conditions (without urgencies) or improving the visualizations of urgency to enhance usability for human users.

## 1. Introduction

Vast amounts of data are being generated daily within healthcare, especially in electronic anesthesia records, where, among other data, continuous patient monitoring data are stored. The ever-increasing use of this data will fundamentally change and improve the way medical care will be practiced in the future [1,2,3,4]. A pressing challenge is to adequately process the data so that caregivers can make evidence-based decisions for the benefit of patients [1]. Machine learning (ML) can curate and analyze large amounts of data, identify the underlying logic, and generate models that can accurately recognize a situation or predict a future state [5,6]. Predictive ML models have already been developed for various fields of medicine [7,8,9]. However, a significant gap exists between the number of developed models, clinically tested applications, and commercially available products [8].

There are several reasons why ML models do not deliver the expected performance in clinical trials [10,11]. One is a lack of trust of the users in the models [7,12,13]. To increase trust, clinically meaningful models should be developed with good unbiased data and should not patronize the users but rather support them in their clinical work [11,14,15,16]. An integral part of such a clinically meaningful model is the presentation of information without imposing an additional cognitive load on the user [17]. A decision support tool that uses a ML model should not lead to alarm fatigue or increased workloads but provide actionable advice that fits into existing workflows [4,13].

To make the ML models that we developed for vital sign predictions in surgical patients clinically meaningful and usable, we developed a user-centered, patient avatar-based graphical representation to visualize vital sign predictions. These visualizations are an extension to Visual Patient (VP), an avatar-based patient monitoring technology [18]. VP has been available in Europe since 2023 as the Philips Visual-Patient-avatar. Studies reported that healthcare providers were able to retrieve more vital signs with higher diagnostic confidence and lower perceived workload when using VP rather than wave- and number-based monitoring, allowing them to obtain a comprehensive picture of the patient’s condition more quickly [18,19]. Additionally, care providers positively reviewed the technology and found it intuitive and easy to learn and use [20].

The project’s objective is to implement vital sign predictions into the VP (provisionally named VP Predictive). To achieve this goal, the project aims to integrate the front-end—i.e., the way predictions are presented to the users—with the back-end—i.e., the ML models calculating the predictions.

In the present study, we report the validation process of the front-end. Specifically, we aimed to determine how accurately users identify the different vital sign prediction visualizations after a short educational video. The development and validation process of the back-end ML models is the subject of a separate study.

## 2. Methods

A declaration of non-jurisdiction (BASEC Nr. Req-2022-00302) was issued by the Cantonal Ethics Committee, Zurich, Switzerland. Due to the study’s exemption from the Human Research Act, ethical approval was not required for the German study centers. Participation was voluntary and without any financial compensation. All participants signed a consent for the use of their data. In reporting the study, we followed the Guidelines for Reporting Simulation Research in Health Care, an extension of the CONSORT and STROBE statements [21].

### 2.1. Study Design and Population

We conducted an investigator-initiated, prospective, multi-center, computer-based simulation study at the University Hospitals of Zurich, Frankfurt, and Wuerzburg. The study consisted of three parts. First, we validated the prediction visualizations by testing their identification by physicians. We included senior and resident physicians employed in the study centers’ anesthesia or intensive care departments according to availability. Following this part, we invited participants from Frankfurt and Wuerzburg to take part in face-to-face, standardized interviews. From the interview transcripts, we identified key topics and derived representative statements. In the third study part, the participants from all three centers rated these statements on Likert scales.

### 2.2. VP and VP Predictive

VP is a user-centered visualization technology specifically developed to improve situation awareness (Appendix A). It creates an animated avatar of the patient to visually display various vital signs according to the real-time conventional monitoring data.

VP Predictive was developed as an add-on to VP, with the goal of integrating vital sign predictions into the standard VP. A prediction consists of a condition and an urgency. The condition signals which vital sign is predicted to change and in which direction (low/high), while the urgency gives the time horizon in which this change is expected to occur. The VP Predictive educational video (Appendix A) and Figure 1 explain the technology.

#### 2.2.1. Condition

There are 22 condition visualizations, which are based on the original VP visualizations. These conditions are displayed as blank visualizations with white dashed borders and superimposed on the VP. The only exception to this display method is oxygen saturation, for which a “low” condition is shown by coloring the blood pressure shadow of the original VP in blue.

#### 2.2.2. Urgency

There are three different urgencies: urgent, intermediate, and non-urgent. For an urgent prediction, the corresponding condition is shown for 3.5 s every 7 s and flashes during the display. An intermediate urgency prediction is shown for 3.5 s every 14 s and does not flash. Finally, a non-urgent prediction is shown for 3.5 s every 28 s and is partially transparent. This way, a more urgent prediction is displayed more frequently than a less urgent one. The additional flashing (urgent) and transparency (non-urgent) are designed to allow users to distinguish the different urgencies upon first viewing.

### 2.3. Study Procedure

We conducted a computer-based simulation study followed by standardized interviews and an online survey.

#### 2.3.1. Part I: Simulation Study

Participants were welcomed into a quiet room. After a short session briefing, and the completion of a sociodemographic survey, we showed the participants a video explaining VP (Appendix A). Afterward, participants had the opportunity to practice on a Philips Visual-Patient-avatar simulator for up to 5 min. Afterward, an educational video explaining VP Predictive was shown (Appendix A).

During the simulation, each participant was shown 33 videos. Each video displayed a standard VP with all vital signs in the normal range, along with an overlaid prediction visualization containing a single condition and urgency. To provide each participant with a randomized set of 33 videos and to ensure that each video was equally represented, we first created randomized sets of 66 videos (3 urgencies × 22 conditions). Then, each set was split in two (videos 1 to 33 and 34 to 66) and watched in sequence by the participants. During the videos, the participants were asked to select the condition shown (22 possible answers) and urgency (3 possible answers). We stopped the video as soon as the participant had answered, or after one minute at the latest. After the participant had completed all questions, we played the next video in the set. All data were collected on an Apple iPad (Apple Inc., Cupertino, CA, USA) using the app iSurvey (Harvestyourdata.org, Wellington, New Zealand) [22].

#### 2.3.2. Part II: Standardized Interviews

After a short break, we conducted a standardized interview with participants from Frankfurt and Wuerzburg. The question was as follows: “What do you think about the VP Predictive visualizations?”. The answers were recorded using an Apple iPhone and later automatically transcribed using Trint (Trint Limited, London, UK). The transcripts were then manually checked for accuracy and translated into English using DeepL (DeepL SE, Cologne, Germany). After manually checking the translation, we divided the text into individual statements for analysis. Using the template approach, we developed a coding tree [23]. Two study authors independently coded each statement. Differences in coding were discussed, and a joint coding per statement was agreed upon.

#### 2.3.3. Part III: Online Survey

Based on the interview results, we created six statements on recurring topics to be rated using Likert scales in an online survey. This survey was designed using Google Forms (Google LLC, Mountain View, CA, USA) and sent by email to all participants of study part I. The survey remained active for three weeks in July–August 2022. Halfway through this period, a single reminder email was sent.

### 2.4. Outcomes

#### 2.4.1. Part I: Simulation Study

We defined *correct prediction identification* as the primary outcome. If participants correctly identified both condition and urgency, we counted this as correctly identifying the prediction. As secondary outcomes, we chose *correct condition identification* and *correct urgency identification*, defined as the correctly identified condition and urgency, respectively. In addition, we analyzed the 22 conditions and the 3 urgencies individually.

#### 2.4.2. Part II and III: Standardized Interviews and Online Survey

For the standardized interviews, we analyzed the distribution of individual statements within the topics of the coding tree. For the online survey, we analyzed the distribution of the answers on the 5-point Likert scale for each statement (from “strongly disagree” to “strongly agree”).

### 2.5. Statistical Analysis

For descriptive statistics, we show medians and interquartile ranges for continuous data and numbers and percentages for categorical data.

#### 2.5.1. Part I: Simulation Study

We used mixed logistic regression models with just an intercept to estimate the correct prediction, condition, and urgency identification while considering that we had repeated, non-independent measurements from each study participant. The estimates are given as percentages with 95% confidence intervals (95% CI). For estimates by condition, we added the condition information to the aforementioned model. We used a mixed logistic regression model to see if there was a learning effect by including the number of the respective question (between 1 and 33). Estimates of this model are given as odds ratios (OR).

#### 2.5.2. Part II and III: Standardized Interviews and Online Survey

In part II of the study, we assessed the agreement of the two coders prior to consensus by calculating the interrater reliability using Cohen’s Kappa. In part III, we used the Wilcoxon matched-pairs signed-rank test to evaluate whether the answers significantly deviated from neutral. We used Microsoft Word, Microsoft Excel version 16.77.1 (Microsoft Corporation, Redmond, WA, USA), and R version 4.2.0 (R Foundation for Statistical Computing, Vienna, Austria) to manage and analyze our data. We used GraphPad Prism version 9.4.1 (GraphPad Software Inc., San Diego, CA, USA) to generate the figures. We considered a *p*-value < 0.05 to be statistically significant.

#### 2.5.3. Sample Size Calculation

To assess the appropriate sample size for the simulation study, we conducted a pilot study with six participants at the University Hospital Zurich. Correct prediction identification was 94.4%. Considering that these participants were already familiar with VP (but did not know VP Predictive), we calculated the sample size based on a true proportion of 90%. In this case, 70 participants are needed to construct a 95% CI for an estimated proportion that extends no more than 10% in either direction.

## 3. Results

We recruited 70 anesthesiologists and intensive care physicians in April–May 2022. All participants completed the simulation study. A total of 21 out of the 70 participants (30.0%) gave an interview, and 49 participants (70.0%) completed the online survey. Table 1 shows the study and participants’ characteristics.

### 3.1. Part I: Simulation Study

#### 3.1.1. Correct Prediction Identification

In total, 1716/2310 (74.3%) prediction visualizations (condition and urgency) were correctly identified. The mixed logistic regression model showed a slightly higher percentage (77.9%, 95% CI 73.2–82.0%).

Figure 2 shows these results for each condition individually. It is apparent that not all conditions were identified equally well. The best-identified conditions showed close to 90% correct prediction identification, whereas a few showed less than 60% correct prediction identification. The mixed logistic regression model-based estimations tended to be a few percentage points higher.

#### 3.1.2. Correct Condition Identification

Considering conditions alone (without urgencies), 2117/2310 (91.7%) were correctly identified. The mixed logistic regression model showed an accuracy of 93.8% (95% CI 93.7–93.8%). Figure 3 shows the correct condition identification for each condition individually. Most conditions were very well identified, with two exceptions: low pulse rate (68.6%) and low respiratory rate (58.1%).

#### 3.1.3. Correct Urgency Identification

Urgency (without condition) was correctly identified in 1855/2310 (80.3%) cases. The mixed logistic regression model accuracy was 84.0% (95% CI 80.2–87.1%). Considering each urgency individually, the urgent one was correctly identified 629/770 (81.7%) times, the intermediate one 577/770 (74.9%) times, and the non-urgent one 649/770 (84.3%) times.

#### 3.1.4. Learning Effect

The mixed logistic regression model showed a significant learning effect on correct prediction identification, with the odds of correctly identifying the predictions increasing by 3% for each additional prediction shown (OR 1.03, 95% CI 1.02–1.04, *p* < 0.001).

### 3.2. Part II: Standardized Interviews

From the transcripts of the interviews, we identified 126 different statements. At first coding, the two independent raters agreed on the classification of 83.3% of the statements (105/126), with a Cohen’s Kappa of 0.8. Most of the positive comments considered VP Predictive to be intuitive. Negative comments mainly concerned identification difficulties, especially with the different urgencies. Several participants noted a learning effect during the session or believed an additional learning effect could be achieved by using VP Predictive more frequently. Figure 4 shows the coding tree in detail. Note that 15.1% of the statements were not codable; these primarily represented statements not relevant to the posed question.

### 3.3. Part III: Online Survey

The questionnaire was completed by 70.0% of the invited participants (49/70). Most of the participants agreed or strongly agreed that VP Predictive was fun to use (32/49, 65.3%) and intuitive (25/49, 51.0%); many of them also agreed or strongly agreed that it was eye-catching (23/49, 46.9%). Almost two-thirds (32/49, 65.3%) agreed or strongly agreed that the urgency identification was difficult. Nevertheless, most participants (31/49, 63.3%) agreed or strongly agreed that they had a steep learning curve during the study session, and only very few (5/49, 10.2%) disagreed or strongly disagreed that they could imagine working with VP Predictive in the future. Figure 5 shows these results in detail.

## 4. Discussion

We sought to investigate VP Predictive. This technology is an extension of the original VP designed to easily represent vital sign predictions with little cognitive load. Participants correctly identified both condition and urgency in the prediction visualizations in almost three quarters of the cases (74.3%). The majority found VP Predictive to be enjoyable to use, with 65.3% rating it as fun and only 16.3% considering it not intuitive.

In this study, correct condition identification was high (91.7%). Regarding the conditions with the lowest percentages of correct identification, i.e., low pulse rate and low respiratory rate, we believe the reason for this result lies in the short display time (3.5 s) combined with the slow movement of the corresponding visualizations. In this short time frame, the visualizations, which move very slowly, perform less than a complete cycle, making users probably less confident about what they saw. We, therefore, believe that a longer display time may solve this problem.

Compared to correct condition identification, correct prediction identification (i.e., correct identification of both condition and urgency) was not an equally high percentage (74.3%). This finding is also in line with the participants’ subjectively perceived difficulty in identifying the different urgencies, as expressed during the interviews and in the survey. The different urgencies aimed to provide vital sign predictions with an expected occurrence time. For example, the prediction for low blood pressure could be displayed with three different urgencies (e.g., 1, 5, or 20 min). The differences in the percentages of correct identification become understandable when considering that the identification of conditions alone involved the interpretation of less visual information than when additional urgencies also needed to be identified.

Interestingly, the primary outcome result in the pilot study differed significantly from the one in the actual study (pilot 94.4% vs. study 74.3% correct prediction identification). One possible explanation for this difference is that the pilot study cohort was already familiar with the original VP visualizations (although not with the prediction visualizations) and, thus, had fewer new things to learn before the study. In comparison, the majority of the actual study participants were encountering VP for the first time. This raises the question of whether a longer familiarization period could have improved the percentage of correct urgency identification, and, thus, also that of correct prediction identification.

This hypothesis is supported by the learning effect that we confirmed quantitatively and from the participants’ feedback. Intuitiveness and learning ease are essential for accepting new technologies and are crucial for their successful clinical introduction [20]. In our case, these requirements seem to have been achieved, as the majority of the survey participants could imagine working with VP Predictive in the future.

Considering our study results, we believe that—with some modifications—VP Predictive may have the potential to display vital sign predictions generated by ML models in a way that healthcare professionals can understand and translate into direct actions. VP Predictive is intended to guide users’ attention. When alerted by a prediction, caregivers should ultimately consider all available information and decide on an appropriate response (e.g., fluids or vasopressors in case of a low blood pressure prediction). However, like any new technology, it needs to be learned and trained before it can be integrated into practice. VPP can probably simplify this process by intuitively displaying predictions in the form of visual representations, compared to using numbers or curves. On the other hand, this can also lead to the loss of potentially useful information.

### Strengths and Limitations

First, the conditions under which the study took place differ from the clinical reality, in which many more factors are present [24]. Second, participants evaluated only videos in which the VP was shown in a physiological state and in which exactly one prediction was shown at a time. Such scenarios differ from the more complex clinical reality, so studies in more realistic settings will be needed to evaluate the true clinical value (e.g., a high-fidelity simulation study) [25]. Third, the standardized interviews were conducted only with willing participants from Frankfurt and Wuerburg, and the online survey was not completed by all participants, thus, reducing the sample size of these two study parts compared to that of the simulation study.

At the same time, a computer-based study also has advantages over a real-life study. First, it allows completely new technologies to be tested without patient risks [26]. It also standardizes the study conditions, an essential prerequisite for minimizing possible bias due to external disturbances.

Another strength of our study is that it was multicenter and multinational, allowing the results to be generalized to a certain extent. Based on the pilot study, the trial was adequately powered; however, the participants’ selection was based on availability during working hours and, therefore, was not random.

## 5. Conclusions

Despite promising results and feedback, the current Visual Patient Predictive visualizations need some modifications followed by further high-fidelity simulation studies to test its suitability for the intended task of displaying vital sign predictions to healthcare providers in an easily understandable way. In this study, care providers correctly identified >90% of the conditions (i.e., without considering urgencies). The percentage of correct identification decreased when considering urgencies in addition to conditions. Therefore, in future development of the technology, we will focus on either only displaying conditions (without urgencies) or on improving the visualizations of urgency to enhance usability for human users.

## Figures and Tables

**Figure 1 diagnostics-13-03281-f001:**
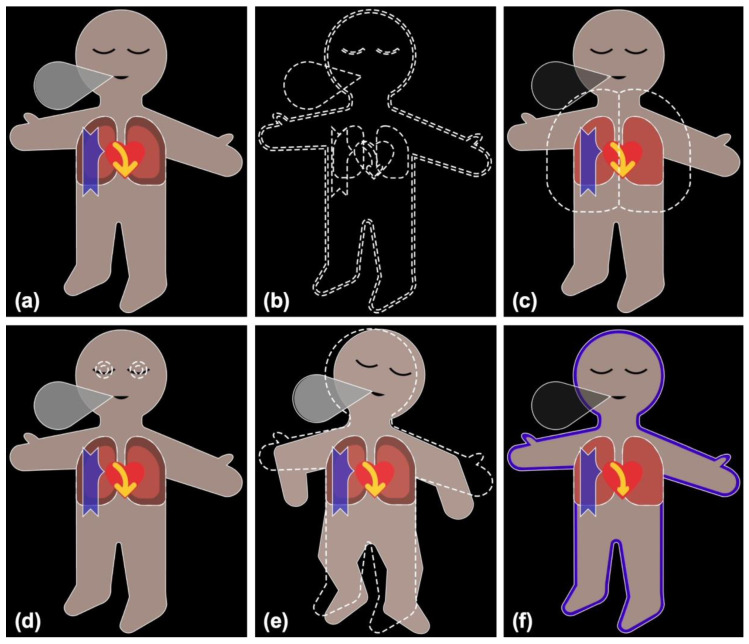
Visual Patient and Visual Patient Predictive. (**a**) Visual Patient displays vital signs in the form of colored visualizations; (**b**) Visual Patient Predictive uses the same visualizations as blank figures with dashed borders. Images (**c**–**f**) show examples where tidal volume (**c**), bispectral index (**d**) and train-of-four ratio (**e**) are predicted to become high, and oxygen saturation (**f**) is predicted to become low, respectively.

**Figure 2 diagnostics-13-03281-f002:**
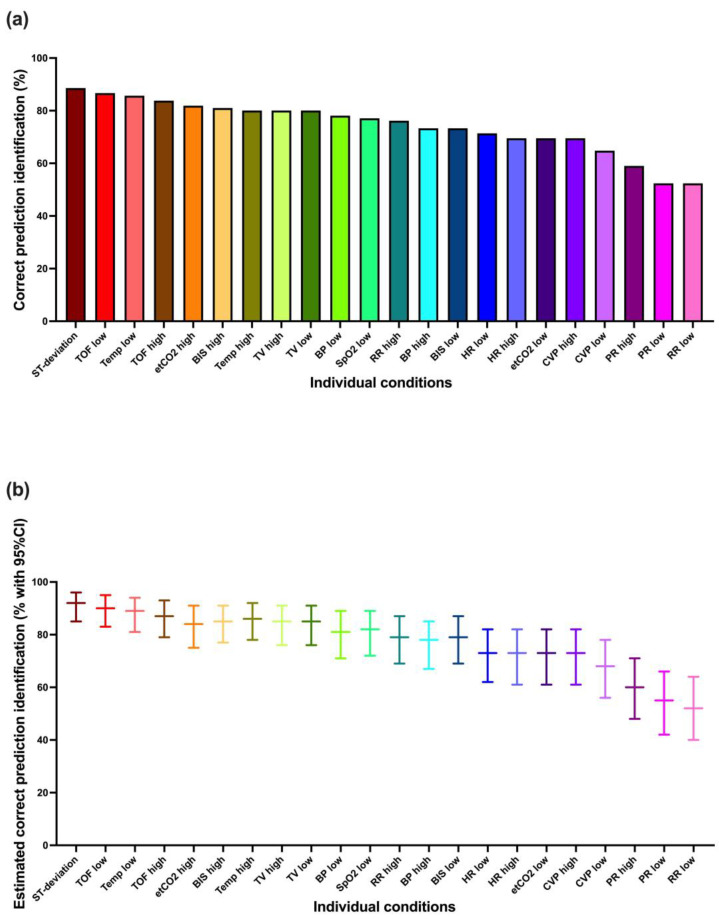
Correct prediction identification (correctly identified condition and urgency) for each condition individually: (**a**) the percentages of correct prediction identification, (**b**) the estimates based on the mixed logistic regression model. ST-deviation, ST-segment deviation; TOF, train-of-four ratio; Temp, body temperature; etCO2, end-expiratory carbon dioxide concentration; BIS, bispectral index; TV, tidal volume; BP, blood pressure; SpO2, oxygen saturation; RR, respiratory rate; HR, heart rate; CVP, central venous pressure; PR, pulse rate (vital signs are color-coded for better readability).

**Figure 3 diagnostics-13-03281-f003:**
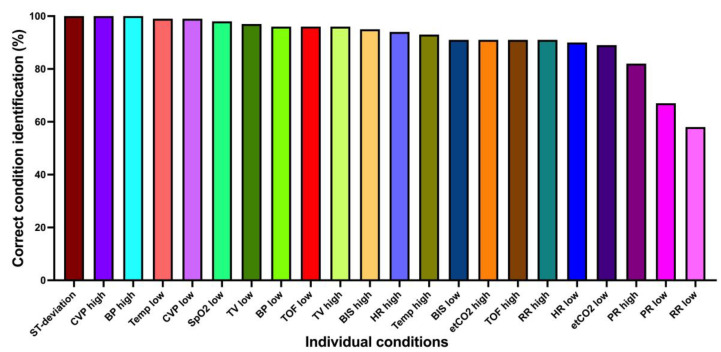
Correct condition identification for each condition individually. ST-deviation, ST-segment deviation; CVP, central venous pressure; BP, blood pressure; Temp, body temperature; SpO2, oxygen saturation; TV, tidal volume; TOF, train-of-four ratio; BIS, bispectral index; HR, heart rate; etCO2, end-expiratory carbon dioxide concentration; RR, respiratory rate; PR, pulse rate (vital signs are color-coded for better readability).

**Figure 4 diagnostics-13-03281-f004:**
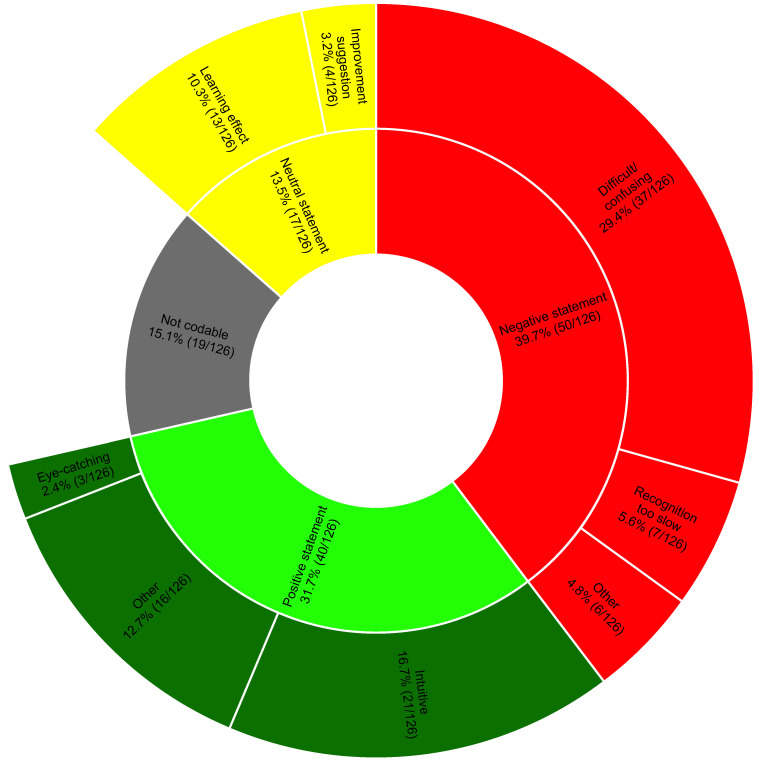
Distribution of the statements within the topics of the coding tree. We show percentages and numbers.

**Figure 5 diagnostics-13-03281-f005:**
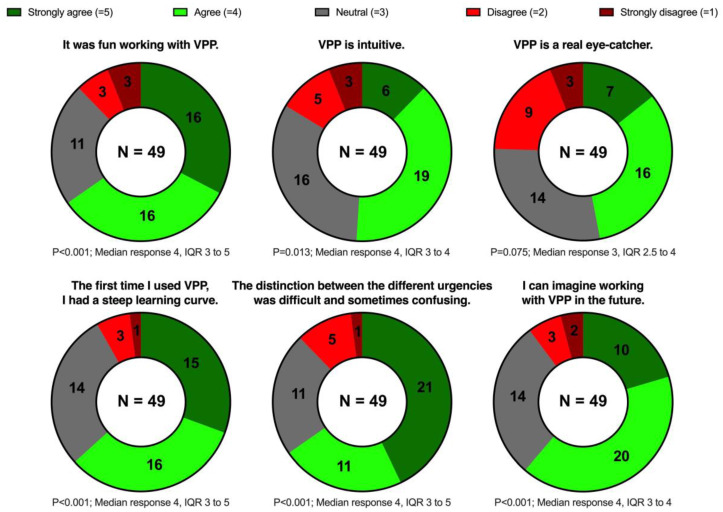
Doughnut charts showing the statements and the distribution of the answers on the 5-point Likert scale. The results are shown as numbers. We calculated *p*-values using the Wilcoxon signed-rank test to determine whether the responses significantly deviated from neutral. VPP, Visual Patient Predictive; IQR, interquartile range.

**Table 1 diagnostics-13-03281-t001:** Participants’ and study characteristics. USZ, University Hospital Zurich; UKW, University Hospital Wuerzburg; KGU, University Hospital Frankfurt; IQR, interquartile range.

	Part I (Simulation Study)	Part II (Standardized Interviews)	Part III (Online Survey)
**Participants characteristics**			
Participants, *n*	70	21	49
Participants from USZ, *n* (%)	35 (50)	0 (0)	
Participants from UKW, *n* (%)	18 (26)	15 (71)	
Participants from KGU, *n* (%)	17 (24)	6 (29)	
Gender female, *n* (%)	42 (60)	15 (71)	
Resident physicians, *n* (%)	56 (80)	17 (81)	34 (69)
Staff physicians, *n* (%)	14 (20)	4 (19)	15 (31)
Age (years), median (IQR)	31 (28–35)	33 (27.5–35.5)	34 (28–37)
Work experience (years), median (IQR)	3.5 (1–6)	3 (1.5–8)	4 (2–7)
Previous experience with Visual Patient, *n* (%)	19 (27)	4 (19)	
**Study characteristics**			
Different conditions studied, *n*	22		
Different urgencies studied, *n*	3		
Different predictions studied, *n*	66		
Randomly selected predictions per participant, *n*	33		

## Data Availability

The data presented in this study are available on request from the corresponding author. The data are not publicly available to protect the privacy of the study participants and ensure they are not identified.

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
