# Peer review of "Using Visual Patient to Show Vital Sign Predictions, a Computer-Based Mixed Quantitative and Qualitative Simulation Study"

_diagnostics, 2023, doi:10.3390/diagnostics13203281_

Round 1

Reviewer 1 Report

The authors conduct a computer-based mixed quantitative and qualitative simulation study using visual patient to show vital sign predictions. Overall, this manuscript is well-written and easy to follow. However, there are some issues that need to be addressed.

What do the different colors mean in Figs 2 and 3?

Why the mixed logistic regression model is used for the identification of prediction visualizations? How about other machine learning approaches?

The references in the last 2 years should be added.

What is the limitation of the proposed method?

Easy to follow.

Author Response

Reviewer 1:

The authors conduct a computer-based mixed quantitative and qualitative simulation study using visual patient to show vital sign predictions. Overall, this manuscript is well-written and easy to follow. However, there are some issues that need to be addressed.

  • What do the different colors mean in Figs 2 and 3?
    • Reply: Thank you for the question. The colors in the figures have been uniquely assigned to the different parameters with the purpose of making them easier to follow and intuitively identify them within Figures 2 and 3. We have now mentioned this in the description of the figures.
    • Change: Pages 7 - 8 (Description Figures 2-3): (Vital signs are color-coded for better readability).
  • Why the mixed logistic regression model is used for the identification of prediction visualizations? How about other machine learning approaches?
    • Reply: Thank you for the input. Before conducting the statistical analysis, we considered the various methods and opted for the mixed logistic regression model, as it is a well-studied and established method for clearly and transparently estimating the odds for binary outcomes with non-independent repeated measures. Machine learning approaches are certainly a valid alternative, but we found them less suitable for a preliminary study such as ours.
  • The references in the last 2 years should be added.
    • Reply: Thank you for the good remark. We have now added more recent references to the manuscript.
    • Change: References 4, 9, 15 and 16 have been added.
  • What is the limitation of the proposed method?
    • Reply: Thank you for the question, which allowed us to improve the quality of the paper. We added a paragraph in the "strengths and limitations" section.
    • Change: Page 11 (lines 316-320): These lines read now: “However, like any new technology, it needs to be learned and trained before it can be integrated into practice. VPP can probably simplify this process by intuitively displaying predictions in the form of visual representations, compared to using numbers or curves. On the other hand, this can also lead to the loss of potentially useful information.”

Reviewer 2 Report

This is interesting study. Manuscript is well written. Study was prospective and multicenter with sample size calculation based on pilot study (but pilot study with low number of participants). It was impossible for me to see attached videos (not found in the system) but I do not think it will change much my review. 

Results of simulation study are very interesting. Concerning lowest correct condition identification rate for PR low and RR low as well as moderate for PR high – does Authors have identified suspected cause of those results and potential method of improvement?

It will be interesting to see the results according to the work experience and previous experience with VP but I understand that sample size may be too low to show these analyses. 

As Authors have mentioned the primary outcome result in the pilot study differed significantly from the one in the actual study what may be explained by the fact that the pilot study cohort was already familiar with the original VP visualizations. How will this influence the learning process int the future? 

The results of standardized interviews and online survey are interesting, but the rate of missing data is relatively high. This should be mentioned in limitation section.

Author Response

This is interesting study. Manuscript is well written. Study was prospective and multicenter with sample size calculation based on pilot study (but pilot study with low number of participants). It was impossible for me to see attached videos (not found in the system) but I do not think it will change much my review. Results of simulation study are very interesting.

Reply: We have re-uploaded the videos and hope you can see them now.

  • Concerning lowest correct condition identification rate for PR low and RR low as well as moderate for PR high – does Authors have identified suspected cause of those results and potential method of improvement?
    • Reply: Thank you for your question, which reinforced the paper. We have integrated this reflection in the discussion.
    • Change: Page 10 (lines 280-286): These lines read now: “Regarding the conditions with the lowest percentages of correct identification, i.e., pulse rate low and respiratory rate low, we believe the reason for this result lies in the short display time (3.5 seconds) combined with the slow movement of the corresponding visualizations. In this short time frame, the visualizations, which move very slowly, perform less than a complete cycle, making users probably less confident about what they saw. We therefore believe that a longer display time may solve this problem.”
  • It will be interesting to see the results according to the work experience and previous experience with VP but I understand that sample size may be too low to show these analyses.
    • Reply: We calculated the sample size for the simulation study according to the primary outcome, i.e., the percentage of correct prediction identification, and conducted the study accordingly. Therefore, the study setting is not suitable for extensive analyses of the different subgroups. Further studies are needed to examine these aspects in detail. In addition, when we conducted the study, VP was not available on the market, so regarding the participants who had previous experience with VP, this derived exclusively from its use in the context of previous studies. In 2023, VP was launched onto the market and is already in use or will be introduced in several institutions. In the future, we will be able to test our new technology also on care providers who use VP daily.
  • As Authors have mentioned the primary outcome result in the pilot study differed significantly from the one in the actual study what may be explained by the fact that the pilot study cohort was already familiar with the original VP visualizations. How will this influence the learning process in the future?
    • Reply: We were very surprised to observe the big differences between the pilot study and the simulation study. We have conducted similar studies for the development of the visual patient and have always found very accurate matches between the pilot and the corresponding study. Our explanation for the discrepancy in this study is described in the discussion (page 10, line 297-302). We learn from this that the conditions for the pilot must match the study to be conducted as closely as possible. Minor details, such as previous experience with the product to be tested, are confounding factors and can bias the results.
  • The results of standardized interviews and online survey are interesting, but the rate of missing data is relatively high. This should be mentioned in limitation section.
    • Reply: Thank you for the good suggestion. We have integrated it into the “strengths and limitations” section.
    • Change: Page 11 (lines 330-334): These lines read now: “Third, the standardized interviews were conducted only with willing participants from Frankfurt and Wuerburg, and the online survey was not completed by all participants, thus reducing the sample size of these two study parts compared to that of the simulation study.”

Reviewer 3 Report

After reading the article "Using Visual Patient to demonstrate vital sign predictions, a computer-based simulation study combining quantitative and qualitative methods," I must say that it leaves a positive impression. The integration of machine learning in the field of medicine is undeniably significant. The manuscript aligns well with the journal's theme and special issue. Additionally, the study design is clearly explained, which is commendable. The study's outcome entailed identifying the challenges that must be tackled for the successful integration of these technologies into medical practice in the future. Thus, I recommend to accept this manuscript in the current form.

Author Response

Reviewer 3:
After reading the article "Using Visual Patient to demonstrate vital sign predictions, a computer-based simulation study combining quantitative and qualitative methods," I must say that it leaves a positive impression. The integration of machine learning in the field of medicine is undeniably significant. The manuscript aligns well with the journal's theme and special issue. Additionally, the study design is clearly explained, which is commendable. The study's outcome entailed identifying the challenges that must be tackled for the successful integration of these technologies into medical practice in the future. Thus, I recommend to accept this manuscript in the current form.

Reply:Thank you for the time you have invested in the manuscript. We are very pleased that you found it interesting and relevant in its current form.

Round 2

Reviewer 1 Report

I have no further comments.